# ReconResNet: Regularised Residual Learning for MR Image Reconstruction of Undersampled Cartesian and Radial Data

**Soumick Chatterjee** [1]                    SOUMICK.CHATTERJEE@OVGU.DE
**Mario Breitkopf** [1]                        MARIO.BREITKOPF@OVGU.DE
**Chompunuch Sarasaen** [1]                    CHOMPUNUCH.SARASAEN@OVGU.DE
**Hadya Yassin** [1]                           HADYA.YASSIN@OVGU.DE
**Georg Rose** [1]                             GEORG.ROSE@OVGU.DE
**Andreas Nürnberger** [1]                     ANDREAS.NUERNBERGER@OVGU.DE
**Oliver Speck** [1]                           OLIVER.SPECK@OVGU.DE
[1] *Otto von Guericke University Magdeburg, Germany*
**Editors:** Under Review for MIDL 2021

## Abstract

MRI is an inherently slow process, which leads to long scan time for high-resolution imaging. The speed of acquisition can be increased by ignoring parts of the data (undersampling). Consequently, this leads to the degradation of image quality. This work proposes a deep learning based MRI reconstruction framework to reconstruct highly undersampled Cartesian or radial MR acquisitions, which includes a modified regularised version of ResNet as the network backbone to remove artefacts from the undersampled image, followed by data consistency steps that fusions the network output with the data already available from undersampled k-space in order to further improve reconstruction quality. The performance of this framework for various undersampling patterns has also been tested, and it has been observed that the framework is robust to deal with various sampling patterns - results in very high quality reconstruction (highest SSIM being 0.990±0.006 for acceleration factor of 3.5), while being compared with the fully sampled reconstruction. It has been shown that the proposed framework can successfully reconstruct even for an acceleration factor of 20 for Cartesian (0.968±0.005) and 17 for radially (0.962±0.012) sampled data.

**Keywords:** MR Image Reconstruction, Radial MRI, Deep Learning, ResNet

## 1. Introduction

Magnetic resonance imaging (MRI) can provide high spatial resolution for detecting minute pathological changes in tissues. However, due to consecutive data acquisition, MRI is an inherently slow process. Fast imaging can improve patient compliance, reduce motion artefacts, increase patient throughput etc. The speed of acquisition can be increased by ignoring parts of the data (k-space), known as undersampling. Taking the inverse Fourier transform from not densely enough sampled k-space frequency data might cause the resultant image to lose resolution and might also have artefacts due to the violation of the Nyquist-Shannon sampling theorem. Many of the approaches that are currently available for the reconstruction of undersampled data are very slow, due to the fact they are very computationally heavy or iterative in nature.

## 2. Methodology

The proposed framework NCC1701 contains two main components: the network backbone architecture - ReconResNet and the data consistency step. This paper proposes ReconResNet with a modified regularised version of the Residual Block (He et al., 2016), by adding a Spatial Dropout between the two convolution layers of the Residual Block. In this model, first, the input is down-sampled with two down-sampling blocks, each decreases the input size by half in all dimensions, while increasing the number of feature maps by two (starting with 64). Next, the network contains 14 modified Residual Blocks. After the residual blocks, the network contains two up-sampling blocks, each doubles the input size and reduces the number of feature maps by half, to obtain the original image size back. Finally, a fully-connected convolution layer is added, followed by Sigmoid as the final activation function. Furthermore, Parametric ReLU (PReLU) has been used as the internal activation function.

The second component of the framework is the data consistency step, which replaces the actual acquired undersampled data in the network's output. In this way, the final output is not entirely depended on the network. The network only helps to fill-in the data, which were ignored during the undersampled acquisition. For undersampled Cartesian data, data consistency was performed following Hyun et al. (2018). For undersampled radial data this paper introduces a new technique where a sampling pattern was generated for a fully sampled image (referred as $\Omega_{FS}$), considered to have twice the number of spokes than its pixel resolution and a NUFFT (Fessler et al., 2007) object was created from it. Then using that object, a forward transform was performed on the output image of the network to obtain its fully sampled radial k-space. The measured spokes were then inserted into the output k-space. A density compensation function (DCF) was applied, followed by adjoint NUFFT using the same NUFFT object was performed to obtain the final output. Only the network backbone is used during the training process. Whereas during inference, the complete framework is used. To train the ReconResNet, the loss has been calculated with the help of the Structural Similarity Index (SSIM), where higher SSIM means closer image similarity. The negative of the SSIM value has been used as the loss value, and it was then be minimised using Adam Optimiser (Initial learning rate 0.0001, decayed by 10 after 50 epochs), and was trained for 100 epochs with a batch size of one. The code of available publicly on GitHub: https://github.com/soumickmj/NCC1701.

Two different datasets were used in this research: OASIS-1 dataset, 150 subjects were used for training and 100 were used for testing, 20% of the training set were used as the validation set and the remaining 80% as the actual training set; and T1 weighted volumes from the IXI-dataset, 100 volumes each were used for training validation and testing. Images from both the datasets were treated as fully sampled images and various undersampled datasets were artificially generated using MRUnder (Chatterjee et al., 2020) pipeline: https://github.com/soumickmj/MRUnder. Artificial Cartesian undersamplings were achieved performed using 1D and 2D variable density masks (Lustig et al., 2007) and radial undersamplings were performed using PyNUFFT (Lin et al., 2018) with the golden angle of 111.25° between the radial acquisitions (spokes).

## 3. Evaluation

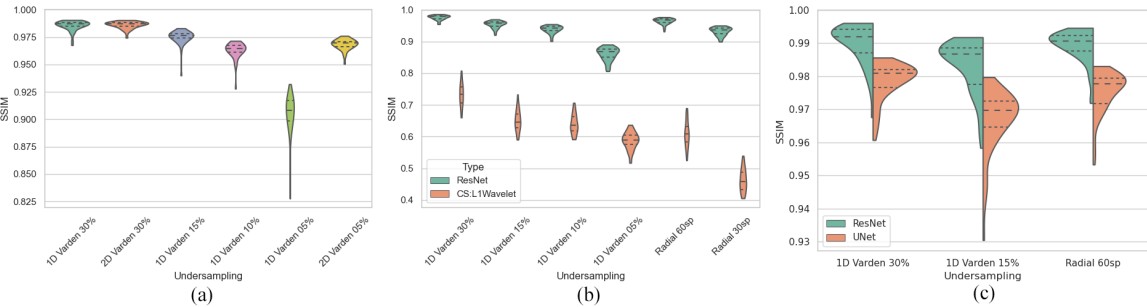

Figure 1: Reconstruction quality (SSIM) of the proposed method for different sampling patterns: (a) OASIS-1 dataset - Cartesian undersamplings, (b) IXI dataset - comparison with L1-wavelet regularised compressed sensing, (c) IXI dataset - comparison with UNet

Different levels of Cartesian undersamplings were evaluated (Fig. 1a) using OASIS-1 dataset: Four levels of 1D Varden (30%, 15%, 10% and 5% of the k-space) and two levels of 2D Varden (30% and 5%). It was observed that the proposed framework could perform successfully even for the highest acceleration factor evaluated in this research, 5% of the k-space - 2D Varden resulted in 0.968±0.005 SSIM, however, 1D Varden resulted in 0.906±0.017. For IXI dataset, four different levels of 1D Varden sampling (30%, 15%, 10%

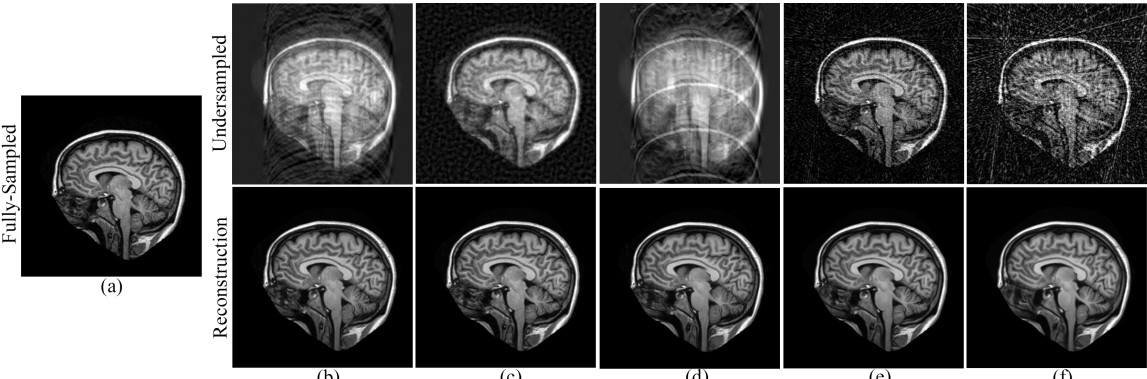

Figure 2: Example output (OASIS-1) of NCC1701: (b) Cartesian 1DVarden30% (c) Cartesian 2DVarden30%, (d) Cartesian Uniform step 4 (e) Radial 60 spokes (f) Radial 30 spokes

and 5% of the k-space) and two different levels of radial sampling (60 and 30 spokes) were evaluated (Fig. 1b) - it can be observed that the proposed method performed significantly better than L1-wavelet regularised compressed sensing reconstruction (Lustig et al., 2007). Moreover, it can be observed (Fig. 1c) that the performance of the proposed method performed significantly better than UNet (Hyun et al., 2018). Independent two-sample t-test has shown that all the observed improvements are statistically significant (p-values always less than $10^{-14}$). Fig. 2 shows an example reconstruction from the OASIS-1 dataset for three different Cartesian and two different Radial undersamplings.

## 4. Conclusion

Evaluation using multiple datasets has shown that the proposed framework can efficiently work with both Cartesian and radial undersampled data, even while trained together, and provides results with high accuracy (SSIM value as high as 0.99) and achieved statistically significant improvements over the baseline methods. Experiments presented here have shown that the framework was able to reconstruct properly for undersampled data with an acceleration factor of 20 for Cartesian (2D Varden 5%) and an acceleration factor of 17 for radial (60 spokes) acquisitions.

### Acknowledgments

This research was supported by the ESF (project no. ZS/2016/08/80646).

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
