# OpenReview forum: "ReconResNet: Regularised Residual Learning for MR Image Reconstruction of Undersampled Cartesian and Radial Data"
_MIDL.io/2021/Conference/Short — MIDL 2021 Poster_

### Official Review · Reviewer_VjU3 · 2021-04-30

**Confidence:** 5
**Final Rating:** 4

**Summary:**

The submitted manuscript presents a timely and interesting reconstruction method which is partly novel. The described achieved acceleration factors (20 for Cartesian, 17 for radial sampling) are impressivly high, in particular because - as not obviously stated differently - data used was based on existing image databases and not complex-valued raw data including multiple channels. Motivation and Methodology are introduced briefly, but appropriate with respect to the MIDl format. Code and relevant references are provided. The evaluation shows clear results also with respect to robustness for a range of (under)sampling patterns. The focus on SSIM as a metric of choice may weaken the overall acceptance and reliability. In the conclusion the authors summarize the achieved reconstruction performance and variability of the proposed (fast) reconstruction method.



**Strengths:**

The authors describe a broad testing strategy with respect to sampling/undersampling schemes. After training the application of the framework (ReconResNet + data consistency) is fast and in combination with the achievable acceleration factors (20 for Cartesian, 17 for radial) a clear benefit to potential applications in MRI. The methods are introduced briefly and references are clear and allow the interested reader to undersand/infrorm about neccessary details. Data were selected from different datasets and results compared to accepted state-of-the-art methods.

**Weaknesses:**

In my opinion the submitted manuscript has some shortcomings and weaknesses, these are mainly in the methology and could have been addressed in a broader discussion instead of the rather short and one-sided conculsion statement.
These are:
- datasets can be investigated, but a short statement about the type of data would have been helpfull. In particular because from an MR-physics perspective the importance of 'true' raw data is recognized.
- data used were real valued images which were handled with a forward model. The transferability of the framework NCC1701 to measured complex-valued, multi-coil data was not addressed or discussed and is potentially not straight forward as dimensionality and data complexity increase drastically.
- Training was optimized based on the same loss function as used for evaluation. At least one other metric such as mean squared error or a visual evaluation in terms of subtraction maps (reference and reconstruction result) would have strengthen this work

**Deanonymize Review:**

yes

**Justification Of The Rating:**

The authors present valuable scientific work and results. Overall the work is described clearly but also briefly with needed references/code given. There are some shortcoming which could have been addressed in a more complete and balance discussion which is missing. However, with respect to current interest of the scientific community to investigate neural networks for reconstruction problems this manuscript shoud be published and therefore given the opportunity to be discussed and developed by a broader audience.

**Paper Type:**

methodological development

**Special Issue:**

yes

---

### Official Review · Reviewer_qAhT · 2021-05-01

**Confidence:** 5
**Final Rating:** 3

**Summary:**

This paper addresses the problem of improving the image quality of MRI Reconstruction. Authors cast their solution using ResNet and data consistency principles.  The experimental results show potentials for different sampling patterns and acceleration factors - this reflects a good initial proof of concept for further testing in more datasets.

**Strengths:**

--  This paper is a short version of a recent paper of the authors.  This paper is well-organised and provides a good view of the motivation. The experiments displayed show potentials and the technique works for different sampling patterns.

**Weaknesses:**

-- Whilst indeed this is a solid short paper, the authors fails in provide the big picture of what are the technical and performance wise advantages of the current model wrt existing DL reconstruction techniques.

-- The description of the technique is indeed enough but there are not clear insight of the optimisation model.

**Deanonymize Review:**

no

**Detailed Comments:**

--This paper is a short version of a recent paper of the authors -- therefore, the authors present a solid experimental results. However, the paper will benefit of having a better description of the technicalities. It would be beneficial to the paper to offer a better picture of the technical contribution and also and interpretation of where the performance gain is coming from.

-- Authors provide a good set of experiments, however, as the space is restricted the discussion on the findings from those experiments (plots and reconstruction outputs) is very limited. Authors should find a good trade-off between the results displayed and the discussion derived from those (the main findings from the experiments).

**Justification Of The Rating:**

This paper is a short version of a recent paper of the authors. Whilst the technical description needs to be better transmitted, the authors provide a good initial proof of concept for improving MRI reconstruction.

**Paper Type:**

both

**Special Issue:**

no

---

### Meta-Review · Area_Chair_dL25 · 2021-05-07

**Recommendation:** Accept (Poster)
**Confidence:** 5

**Metareview:**

The paper is judged favorably by both reviewers and fits well within MIDL's concept of allowing recently published journal papers to be discussed at the conference.

---

### Decision · Program_Chairs · 2021-05-11

Accept (Poster)